# The Impact of Protein Glycosylation on the Identification of Patients with Pediatric Appendicitis

**DOI:** 10.3390/ijms25126432

**Published:** 2024-06-11

**Authors:** Dalma Dojcsák, Flóra Farkas, Tamás Farkas, János Papp, Attila Garami, Béla Viskolcz, Csaba Váradi

**Affiliations:** 1Advanced Materials and Intelligent Technologies Higher Education and Industrial Cooperation Centre, University of Miskolc, 3515 Miskolc, Hungary; dalma.dojcsak@uni-miskolc.hu (D.D.); bela.viskolcz@uni-miskolc.hu (B.V.); 2Borsod-Abaúj-Zemplén County Center Hospital and University Teaching Hospital, 3526 Miskolc, Hungary; vfflora@bazmkorhaz.hu (F.F.); gyekseb@bazmkorhaz.hu (T.F.); drpaja.mail@gmail.com (J.P.); 3Institute of Energy, Ceramic and Polymer Technology, University of Miskolc, 3515 Miskolc, Hungary; attila.garami@uni-miskolc.hu; 4Institute of Chemistry, Faculty of Materials Science and Engineering, University of Miskolc, 3515 Miskolc, Hungary

**Keywords:** serum glycosylation, liquid chromatography, mass spectrometry, appendicitis

## Abstract

The identification of pediatric appendicitis is challenging due to the lack of specific markers thereby several factors are included in the diagnostic process such as abdominal pain, ultrasonography and altered laboratory parameters (C reactive protein, absolute neutrophil cell number and white blood cell number). The glycosylation pattern of serum N-glycome was analyzed in this study of 38 controls and 40 patients with pediatric appendicitis. The glycans were released by enzymatic deglycosylation followed by fluorescent labeling and solid-phase extraction. The prepared samples were analyzed by hydrophilic interaction liquid chromatography with fluorescence and mass-spectrometric detection. The generated data were analyzed by multiple statistical tests involving the most important laboratory parameters as well. Significant differences associated with the examined patient groups were revealed suggesting the potential use of glycosylation analysis supporting the detection of pediatric appendicitis.

## 1. Introduction

Appendicitis is one of the most common causes of surgical procedures of the acute abdomen where 96.5–100/1000 cases are acute inflamed appendix resulting in abdominal surgery [1]. The number of cases of appendicitis are also increased in industrialized countries meaning a main surgical problem, affecting both the elderly and children [2,3]. Based on estimated data, 30% of children presenting abdominal pain are diagnosed with acute appendicitis [4,5].

The main problem with acute appendicitis is the lack of decisive testing to confirm the diagnosis, often resulting in unnecessary surgical intervention [6]. The most common symptoms of appendicitis include dull periumbilical pain, nausea, loss of appetite, migration of pain to the right lower quadrant and low-grade fever [1,7]. In general, clinical diagnosis of acute appendicitis can be confirmed based on clinical history, physical examination, blood tests and imaging diagnostics, such as ultrasound, computed tomography and magnetic resonance imaging [8]. In spite of the diagnostic algorithm supplemented by appendicitis scoring systems, the final decision whether to perform a surgery could be challenging in several cases, especially in children [9]. It is important to note that difficulty in communication with children may result in missing data on their complaints and during the abdominal examination. Moreover, the classic symptoms are often missed in young children [10]. In addition, assertively obtaining diagnostic images using US examination could be difficult in obese patients, due to meteorism and the unusual position of appendix [11].

One of the main markers in establishing the diagnosis of appendicitis is the C reactive protein (CRP) level which is often related to the severity of the disease. The elevated level of procalcitonin (the precursor of calcitonin) can reportedly influence surgical decisions [12], while the bilirubin level was also found as a useful predictor of appendicular perforation [7,13]. A higher level of leucine-rich alpha-2-glycoprotein 1 in saliva was observed in patients with acute appendicitis by enzyme-linked immunosorbent assay, with high specificity and sensitivity to appendicitis according to receiver operating characteristic (ROC) analysis [14]. Similarly to the above-mentioned parameters, the determination of novel specific biomarkers could improve the identification of acute appendicitis and the differentiation in comparison to abdominal pain [15,16].

Protein glycosylation is a critical post-translational modification reportedly altered in a wide range of pathological conditions including cancer and inflammation [17,18]. Inflammatory immune responses are the mixtures of altered systemic physiological and biochemical processes. Pro-inflammatory cytokines can modulate the expression levels of glycosyl-transferases impacting on the biosynthesis of glycan chains resulting in altered glycosylation patterns of parent proteins [19]. The importance of sialylation, fucosylation, galactosylation and terminal bisecting N-acetyl-glucosamine is identified in multiple immune reactions such as antibody-dependent cell-mediated cytotoxicity, complement activation and affinity to Fcγ receptors [20]. Alterations of N-glycosylation were found to be associated with metabolic health, inflammatory markers and correlating with CRP [21]. The analysis of the altered glycosylation patterns may improve the detection of inflammation in combination with current diagnostic methods [17]. Due to the complexity of glycan structures, their analysis requires high-resolution separation methods mostly liquid chromatography and capillary electrophoresis with fluorescence or mass-spectrometric detection [22,23].

In this study, our goal was to reveal potential glycosylation-based differences in childhood appendicitis compared to healthy controls in order to support diagnostic decision-making. Glycans were released from serum samples by PNGase F digestion followed by fluorescent labeling and solid-phase extraction. The prepared samples were analyzed by HILIC-FLR-MS in order to quantify the individual glycan structures. Multiple statistical tests were performed on the generated N-glycomic dataset in combination with the most decisive laboratory parameters suggesting clear correlations regarding the patient groups.

## 2. Results and Discussion

The aim of this study was to investigate the alterations of serum N-glycosylation in childhood patients diagnosed with appendicitis. In total, 38 control and 40 serum samples diagnosed with appendicitis were analyzed by HILIC-FLR-MS in triplicates as is shown in Figure 1. As a result of total ion chromatograms, each glycan structure was identified by its measured mass-to-charge (*m*/*z*) values (Appendix A). The relative quantitation of the individual glycan structures was performed using the fluorescence chromatograms, where 48 peaks were integrated which were used for statistical analysis. The average area% data of each identified glycan can be found in Appendix A.

Integrated data were used for statistical tests to find the correlation and significant differences between the control and appendicitis sample groups. Before examining the correlation, a numerical dataset analysis was performed on the control and patient groups, which were plotted as distribution curves and boxplot diagrams. The boxplot chart was created to display a summary of a set of data values with minimum, first quartile, median, third quartile and maximum properties. Distribution curves showed that in most cases, our data cannot be described by normal distribution (Appendix A). As a confirmation, normality testing was performed by Shapiro–Wilk and D’Agostino’s K^2^ tests with a 95% confidence interval suggesting that most of the data did not follow the normal distribution. In order to find a correlation between the relative amount of distinct N-glycan structures and appendicitis, Spearman rank correlation was applied. It is important to note that for the correlation test, the structures with a value of area% below 0.4 were not included in the statistics. Inflammatory laboratory data such as CRP, absolute neutrophil cell number (ANC) and white blood cell number (WBC) were also included in the statistical analysis as independent variables (original data shown in Appendix A). The correlation coefficients were labelled by color and ranged from 0 to 10 as is shown in the heatmap of Figure 2. To find the correlation between glycan structures (independent variable) and disease scores (discrete variable), Point-Biserial correlation analysis was applied, as is shown in Appendix A.

Several glycan structures (A2G2S1#19, A2G2S2#25, A2BG3S2#31 and A3G3S3#34) describe the presence of the disease (labelled disease score on the heatmap) by the correlation value (R) between 7 and 10, which means a strong positive linear relationship. This result was derived from the Point-Biserial test, as is shown in Table 1. Furthermore, it was observed that these structures also have a strong relationship (R ≥ 7) with other glycan structures based on the heatmap and resulting data of the Spearman correlation from Table 1. Moreover, the high correlation (R = 10) between ANC and WBC laboratory data was also observed by the Spearman test (Table 1).

To identify significant alterations between control and patient groups, the Kruskal–Wallis test has been accomplished with a 99% confidence interval and 0.01 significance level (Appendix A). Similarly, to the results of the heatmap in Figure 2, significantly different structures were the same four glycan structures (A2G2S1#19, A2G2S2#25, A2BG3S2#31 and A3G3S3#34).

The pairplot diagrams are presented in Figure 3, suggesting clear separation of the control and appendicitis groups based on the area% of A2G2S1#19, A2BG3S2#31 and A3G3S3#34 glycan structures. The correlation between A2G2S1#19, A2BG3S2#31 and A3G3S3#34 glycan structures with other glycans mainly affected the degree of the separation of the patient groups. The higher level of A2G2S1#19 structure clearly defines the control group depending on the area% of A2FG2S1#20 or A3G3S3#34, while overlaps can be observed depending on the area% of A2G2S2#25 and A2BG3S2#29.

Overall, the lower level of neutral glycans and the higher area % of sialylated structures mainly contributed to the clear separation of the appendicitis and the control groups. A2G2S1#19 and A2BG3S2#31 levels were decreased, while A3G3S3#34 level was increased in the appendicitis group. It can be also observed that the patient groups were clearly identified when the area level of A2G2S1#19 was higher than 0.4%. A similar pattern of separation was observed regarding the CRP value, as the appendicitis group could be defined by the decreased level of A2G2S1#19 and higher CRP values (Appendix A). The glycans contributing to the separation between groups were also confirmed by ROC analysis, as shown in Figure 4. The data plotted on the ROC curve are more representative of the given classification criterion when the larger the area under the curve (AUC), 0.85 AUC value was chosen as the limit for the reliability of the test and represented only structures with AUC values above 0.85.

As a result of ROC analysis (Table 2), the control group was characterized by A2G2S1#19 and A2BG3S2#31 structures with a sensitivity above 90%. The appendicitis patient group could be defined by the A3G3S3#34 structure with a 94% sensitivity, while in the case of the A2G2S2#25 with 79% sensitivity. The specificity values were 100%, except A2G2S2#25 (99%), and the significance levels were below 0.001 in each cases.

Based on the ROC analysis and Kruskal–Wallis test, four glycan structures were identified with the following assumptions.

If the A2G2S1#19 structure has a relative area% above 0.44, the sample could be classified into the control group with 94% sensitivity and 100% specificity (Appendix A).If the A2BG3S2#31 structure has a relative area% above 0.84, the sample could be classified into the control group with 91% sensitivity and 100% specificity (Appendix A).If the A3G3S3#34 structure has a relative area% above 0.25, the sample could be classified into the appendicitis patient group with 94% sensitivity and 100% specificity (Appendix A).If the A2G2S2#25 structure has a relative area% above 25, the sample could be classified into the appendicitis patient group with 79% sensitivity and 99% specificity (Appendix A).

The assumptions were confirmed by the scatter plots shown in Appendix A and the summary data shown in Appendix A.

The diagnostic efficiency in the case of glycan biomarkers was also verified by the examination of the abdominal pain control subgroup. For the statistic, ROC analysis was used (Appendix A) and the significance results were confirmed by the Kruskal–Wallis test and visualized in the scatter plots of Figure 5. Similarly, to our previous results, the abdominal pain control group was separated by the A2G2S1#19 and A2BG3S2#31 glycan structures with high sensitivity (A2G2S1#19: 100%, A2BG3S2#31: 91%) and specificity (A2G2S1#19, A2BG3S2#31: 100%) compared to the group diagnosed with appendicitis. In addition, the appendicitis patient group could be distinguished based on the laboratory data and the level of A3G3S3#34 glycan structure with high sensitivity (A3G3S3#34: 98%, CRP: 86%, ANC: 82%, WBC: 80%,) and specificity (A3G3S3#34: 100%, CRP: 100%, ANC: 100%, WBC: 100%,) compared to the abdominal pain control group (Appendix A). In the case of these structures, average area% values were determined, which could classify the samples into the diseased or the control groups. If the average area% of A2G2S1#19 is above 0.44 and average area% of A3G3S3#34 structure is below 0.25, the sample is classified into an appendicitis patient group. Observations of the glycan structures provided with 91% sensitivity and 100% specificity are visualized as scatter plot diagrams in Figure 5.

The importance of glycosylation analysis in the identification of pediatric appendicitis was revealed in this study. Our results demonstrated a strong correlation of individual glycan structures (A2G2S1#19, A2G2S2#25, A2BG3S2#31 and A3G3S3#34) with the appearance of acute appendicitis and the level of CRP value. The evidence of a higher level of sialylation indicating a decrease of neutral structures was presented by multiple statistical approaches (Figure 5, Appendix A). The relative area percentages were determined by the significantly different glycan levels, in order to classify the control and appendicitis patient groups. The control group was also divided into normal and abdominal pain control groups, where the level of A2G2S1#19 was significantly lower in appendicitis patients compared to abdominal pain and normal control groups, while the A2G3S3#34 was found to be higher. Our results are in perfect agreement with previous studies where higher sialylation was found to be associated with inflammation [19,21].

## 3. Materials and Methods

### 3.1. Chemicals

Procainamide hydrochloride and dimethyl sulfoxide, formic acid, acetic acid, acetonitrile and picoline borane, were obtained from Sigma-Aldrich (St. Louis, MO, USA). Ammonia solution was purchased from Scharlab S.L. (Barcelona, Spain). PNGase F was provided by New England Biolabs (Ipswich, MA, USA).

### 3.2. Patient Samples

Controls and patients were admitted to the Department of Pediatric Surgery in Miskolc, Hungary between March 2021 and January 2023. In this study, we have selected patients who were less than 18 years old and had no previous history of any chronic disease receiving special therapy. We have distinguished two main groups, including patients with the diagnosis of acute appendicitis and the control group. Within the controls, a subgroup with abdominal pain was also separated as a false positive group.

-Control group

The majority of the patients in the control group required traumatological care for their different types of wounds, fractures and injuries, such as concussion. Patients with the diagnosis of any inflammatory or infectious disease were excluded and whose CRP level was more than 10 mg/L, which is the upper limit of normal based on the local laboratory standard. In this group, we have represented 38 samples.

-Abdominal pain control subgroup

The abdominal pain control group included samples from children who were admitted with abdominal pain, but normal CRP and no confirmation of acute appendicitis. This group represented nine serum samples within the control sample group.

-Patients with acute appendicitis

Each patient with the suspicion of acute appendicitis underwent (1) physical examination (looking for direct and indirect signs of appendicitis), (2) abdominal ultrasound (searching for predictors of appendicitis, such as enlarged appendix, peri-appendiceal fat stranding, appendiceal wall thickening and appendiceal wall enhancement), (3) blood test (complete blood count (CBC), inflammatory markers (CRP and procalcitonin if needed), liver and kidney function tests) and (4) urinalysis before the surgical procedure. The final diagnosis was completed with the histological results. In this group, we have collected 40 samples. The parents have signed a general consent form in accordance with the Declaration of Helsinki when children were admitted. The study was approved by the Ethics and Medical Research Committee of Borsod-Abaúj-Zemplén County Central Hospital and University Teaching Hospital.

### 3.3. N-Glycan Release from Serum Proteins, Labelling and Clean Up

An amount of 9 µL of serum sample was deglycosylated by the PNGase F digestion protocol of New England Biolabs (Ipswich, MA, USA). The released glycans were fluorescently derivatized by the addition of 10 µL 0.3 M procainamide and 300 mM picoline borane in 70%/30% of dimethyl sulfoxide/acetic acid and incubated for 4 h at 65 °C. The sample clean-up was performed by NH_2_-functionalized MonoSpin columns (GL Sciences Inc., Tokyo, Japan) based on the manufacturer’s protocol. The purified glycans were dissolved in 25%/75% water/acetonitrile and separated by liquid chromatography.

### 3.4. UPLC-MS Analysis

The fluorescently labeled and purified N-glycans were analyzed by a Waters Acquity ultra-performance liquid chromatography instrument equipped with a fluorescence detector and a Xevo-G2-XS QToF mass spectrometer. The LC-MS system was controlled by MassLynx 4.2 (Waters, Milford, MA, USA). Waters BEH Glycan column, 100 × 2.1 mm i.d., 1.7 µm particles was used by a linear gradient of 25–45% 50 mM ammonium formate pH 4.4 (Buffer A) at 0.4 mL/min in 42 min, using acetonitrile as Buffer B. A 5 µL sample was injected in partial loop mode in all separations. The temperature of the sample manager was set at 15 °C, while the column temperature was 60 °C during each run. The fluorescence excitation was λ_ex_ = 308 nm and the emission was λ_em_ = 359 nm. In the MS, the applied electrospray voltage was 2.2 kV while the desolvation temperature was 120 °C with the desolvation gas flow of 800 L/h. Mass spectra were monitored in positive ionization between 500 and 2000 m/z. MS/MS fragments were obtained using a ramp voltage between 30 and 60 kV collision energy to obtain the fragmentation data of the separated glycan structures (Appendix A).

### 3.5. Data Analysis

Python, IBM SPSS Statistics Version 20 (Armonk, New York, USA) and GraphPad Prism 10.1.2. (Boston, MA, USA) software were used for the tests and visualization. Shapiro–Wilk and D’Agostino’s K^2^ tests were used to test the normality of continuous variables. The tests were performed with a 95% confidence level. The Kruskal–Wallis test was implemented to look for significant changes in glycan structures between control and patient groups. The test was performed with a 99% confidence interval and 0.01 significance level. Visualization of the relationship between each variables was performed by pairplot diagrams [24]. Specific structures were also confirmed by receiver operating characteristic (ROC) between the control and patient groups. ROC analysis was performed with a 99% confidence level by the software. GlycoWorkbench 2.0 was used to identify the analyzed N-glycan structures based on their mass to charge ratio.

## Figures and Tables

**Figure 1 ijms-25-06432-f001:**
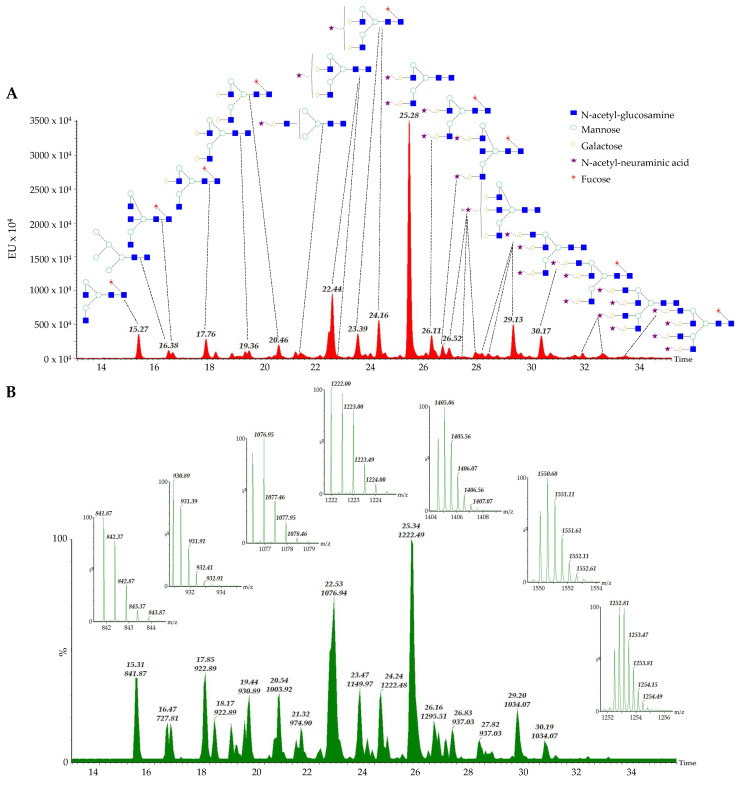
Representative fluorescence (**A**) and total ion chromatogram (**B**) of serum N-glycome from an appendicitis patient by HILIC-FLR-MS.

**Figure 2 ijms-25-06432-f002:**
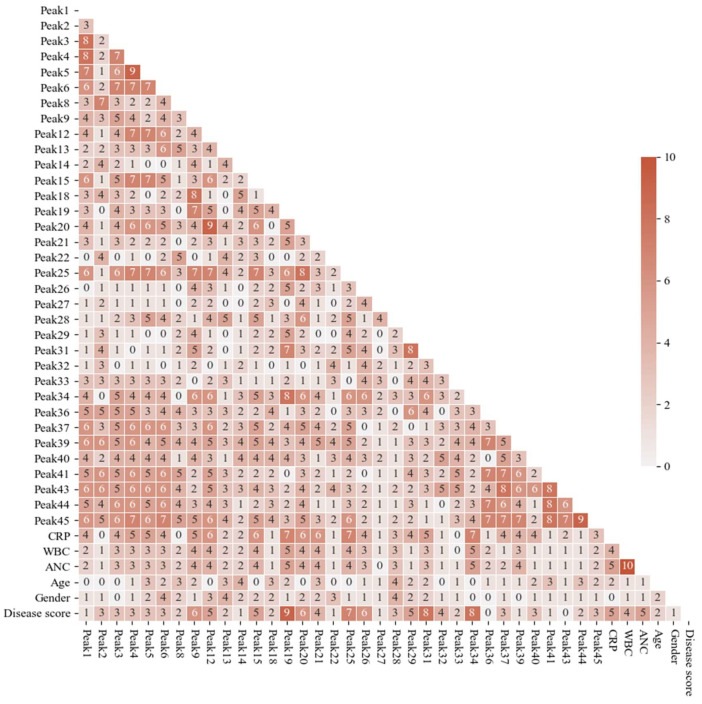
Heatmap of Spearman and Point-Biserial correlation analysis representing the impact of individual glycan peaks on the appendicitis disease score. Values higher than 7 were considered as strong correlations. (the correlation coefficient values of the parameter pairs are interpreted as 10^−1^).

**Figure 3 ijms-25-06432-f003:**
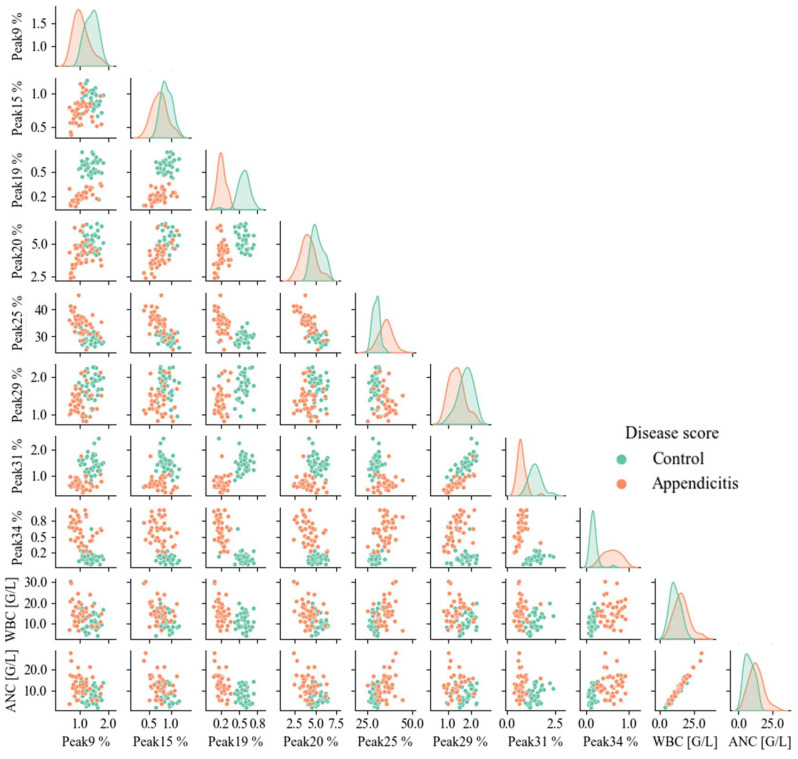
Pairplot diagrams of significantly different glycan structures depending on the disease score.

**Figure 4 ijms-25-06432-f004:**
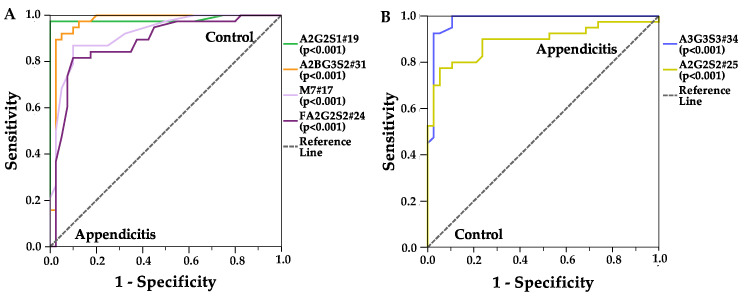
ROC curves for the glycan peaks which significantly changed in control (**A**) or appendicitis groups (**B**). Null hypothesis: true area = 0.5. Confidence level was 99%.

**Figure 5 ijms-25-06432-f005:**
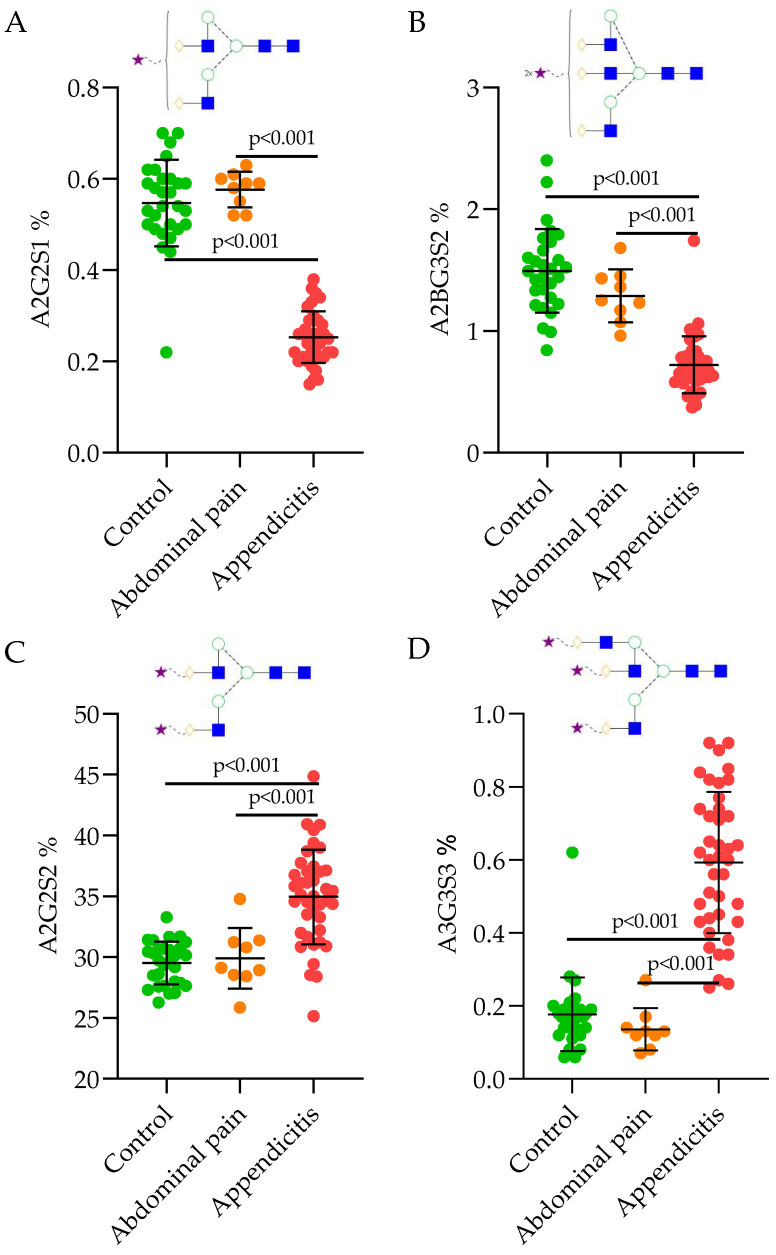
Significantly different glycan structure (A2G2S1 (**A**), A2BG3S2 (**B**), A2G2S2 (**C**), A3G3S3 (**D**)) levels in the controls compared to the group with abdominal pain and appendicitis.

**Table 1 ijms-25-06432-t001:** Correlation of results according to the heatmap.

Glycan Structure and Disease Score Correlation (Point-Biserial Test: *** *p* < 0.001)	Other Glycan Structures Correlations (Spearman Test: *** *p* < 0.001)
*** A2G2S1#19	*** A2G2#9
	*** A2BG3S2#31
	*** A3G3S3#34
*** A2G2S2#25	*** FA2G1#4
	*** FA2G1#5
	*** A2G2#9
	*** FA2G2#12
	*** M4G1S1#15
	*** A2FG2S1#20
*** A2BG3S2#31	*** A2BG3S2#29
	*** A2G2S1#19
*** A3G3S3#34	*** A2G2S1#19
**Laboratory data correlation with other data** **(Spearman test: *** *p* < 0.001)**	
CRP	*** A2G2S1#19
	*** A2G2S2#25
	*** A2BG3S2#31
WBC	*** ANC

**Table 2 ijms-25-06432-t002:** The respective AUC, sensitivity and specificity results of ROC analysis. The sensitivity refers to the number of real positive tests divided by all the real positive and false negative tests. The specificity is determined by the number of real negative tests divided by the total real negative and false positive tests. (A2G2S1#19, A2BG3S2#31 and M7#17 define the control while FA2G2S2#24, A3G3S3#34 and A2G2S2#25 identify the appendicitis group).

Test Variables	AUC	Std. Error	Asymptotic 99% Confidence Interval	
			Sensitivity	Specificity
A2G2S1#19	0.98	0.02	94%	100%
A2BG3S2#31	0.97	0.02	91%	100%
M7#17	0.92	0.03	84%	99%
FA2G2S2#24	0.88	0.04	78%	97%
A3G3S3#34	0.98	0.02	94%	100%
A2G2S2#25	0.89	0.04	79%	99%

## Data Availability

The generated data can be requested from the corresponding author.

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
