# Peer review of "The Impact of Protein Glycosylation on the Identification of Patients with Pediatric Appendicitis"

_ijms, 2024, doi:10.3390/ijms25126432_

Round 1

Reviewer 1 Report

Comments and Suggestions for Authors

The authors describe investigations designed to examine whether pediatric appendicitis is correlated with alterations of the serum N-glycome.

To do so, the serum N-glycome was analyzed by liquid chromatography coupled to mass spectrometry. Differences in the relative quantity of N-glycome structures were statistically evaluated. The results indicate an increase of (tri)sialylated complex N-glycans in the serum of pediatric patients with appendicitis what is in line with many previous findings of a increase of sialylted N-glycans in inflammation.

The manuscript might be of interest. However, it has severe deficits that rule out publication in the present form.

1.     Throughout. The manuscript is poorly written with numerous mistakes in grammar and phrasing.

2.     Introduction. The selection of references is random and important references that represent the field of research are missing.

3.     Results and Discussion. The description of the procedures and of the data is in larger parts of the manuscript scarce and cursory rendering the manuscript partly incomprehensible. For instance:

-          Figure 1. The legend is insufficient. Hence, one cannot understand which samples have been analysed. Controls or patients´samples?

-          Line 89. What is meant with the phrase “two patients groups”? According to the Material and methods section there was one patient group.

-          Table 1/Table S1. The authors must describe how peaks were assigned to N-glycan structures.

-          Figure 2. Again, the legend is totally insufficient. What is ANC? Please, explain abbreviations.

-          Lines 131/132. The phrasing of these sentences leaves it open which type of N-glycan alterations correlates with either controls or the patient group.

-          Table 2. The authors must define sensitivity and specificity with respect to the control and the disease group, respectively.

-          Line 170/lines 221 ff. The definition of the abdominal pain group remains unclear. According to line 223 the children belonging to this group were diagnosed with acute appendicitis despite normal CRP concentrations. If this is the case, and since these 9 patients were assigned to the control group, the control group would include patients with acute appendicitis and normal CRP. Since the time-course of CRP and N-glycome alterations may be different, this would question the validity of the control group.

-          Control/patients group. The authors must present a table summarizing the principle data of the patients (e.g. age, gender, CRP and WBC counts, pain etc.)

-          Lines 233-236. Accordingly, ethics and informed consent were solely valid for the patients´group. What about the control group?

-          Table S1. No legend! Does the average area% represent patients or controls? Data of the two groups must be given.

-          Table S2. No legend!

-          Figure S1. Again, this figure is not comprehensible. The legend does not explain what is presented. What is the meaning of A,B,C,D?

-          Figure S2. Dito

-          Figure S5. What does “lower level” mean?

Comments on the Quality of English Language

Author Response

Dear Reviewer,

Please find our response in the attached pdf.

Reviewer 2 Report

Comments and Suggestions for Authors

The authors analyzed N-glycans derived from the serum of healthy pediatric controls, pediatric controls suffering abdominal pain, and pediatric patients with appendicitis. Several N-glycans of individual mass are quantitated and used for comparing the three groups through statistical tests.

The topic of the article is interesting, mass spectrometry and statistical analyses are well performed. The authors claim to distinguish between neutral and sialylated peaks but there is no characterization of any peak. I strongly recommend analyzing the samples before and after sialidase treatment before drawing conclusions.

Changes of the N-glycome of serum glycoproteins during inflammation are well known. I suggest using another control group comprising pediatric patients suffering other acute inflammation processes not related to appendicitis. I think that such additional control group is necessary to corroborate the conclusions.

Author Response

Dear Reviewer,

Thank you for your suggestions.

Please find our response in the attached pdf.

Round 2

Reviewer 1 Report

Comments and Suggestions for Authors

The authors have significantly improved the manuscript, particularly in the Supplementary Material section. Nevertheless, there are still some deficits that can and should further be improved.

-          Figure 1. The legend is still insufficient. The authors must describe that a sample taken from an appendicitis patient has been analysed. The purpose of the legends to figures and tables is to provide the reader with clearly understandable information about the content so that he or she can understand the content unambiguously.

-          Line 100. The phase “glycosylation level” is slang and is not suited to describe what is meant. What is meant is the relative amount of distinct N-glycan structures in the control as compared to the patients´ group.

-          Figure 2. Again, the legend is still insufficient.

-          Table 2. The definition how “sensitivity” and “specificity” were used should be explained in some more detail.

Comments on the Quality of English Language

Quality of English language is now appropriate

Author Response

The authors have significantly improved the manuscript, particularly in the Supplementary Material section. Nevertheless, there are still some deficits that can and should further be improved.

-          Figure 1. The legend is still insufficient. The authors must describe that a sample taken from an appendicitis patient has been analysed. The purpose of the legends to figures and tables is to provide the reader with clearly understandable information about the content so that he or she can understand the content unambiguously.

Figure 1 legend has been corrected.

-          Line 100. The phase “glycosylation level” is slang and is not suited to describe what is meant. What is meant is the relative amount of distinct N-glycan structures in the control as compared to the patients´ group.

Thank you for your recommendation. We have modified based on your suggestion.

-          Figure 2. Again, the legend is still insufficient.

The legend of Figure 2 has been corrected.

-          Table 2. The definition how “sensitivity” and “specificity” were used should be explained in some more detail.

Thank you for your comment. The explanation has been added to Table 2.

Reviewer 2 Report

Comments and Suggestions for Authors

The authors did not follow the recommendations. The revision is substantially identical to the previous  version, so my evaluations.

Author Response

Thank you for your recommendations. We have attached our replies.
